Structures and functions of invertebrate
glycosylation. *Open Biol.* **9**: 180232.

**Subject Area:**
molecular biology

glycosylation, invertebrate, glycoconjugates,
structure and function, glycan profiling

**Author for correspondence:**
Keping Chen
e-mail: kpchen@ujs.edu.cn

# Structures and functions of invertebrate glycosylation

Feifei Zhu[1,2], Dong Li[1] and Keping Chen[1]

[1]Institute of Life Sciences, and [2]School of Food and Biological Engineering, Jiangsu University, Zhenjiang 212013, People's Republic of China

(iD) FZ, 0000-0003-4152-0857; KC, 0000-0001-5254-2299

Glycosylation refers to the covalent attachment of sugar residues to a protein or lipid, and the biological importance of this modification has been widely recognized. While glycosylation in mammals is being extensively investigated, lower level animals such as invertebrates have not been adequately interrogated for their glycosylation. The rich diversity of invertebrate species, the increased database of sequenced invertebrate genomes and the time and cost efficiency of raising and experimenting on these species have enabled a handful of the species to become excellent model organisms, which have been successfully used as tools for probing various biologically interesting problems. Investigation on invertebrate glycosylation, especially on model organisms, not only expands the structural and functional knowledgebase, but also can facilitate deeper understanding on the biological functions of glycosylation in higher organisms. Here, we reviewed the research advances in invertebrate glycosylation, including N- and O-glycosylation, glycosphingolipids and glycosaminoglycans. The aspects of glycan biosynthesis, structures and functions are discussed, with a focus on the model organisms *Drosophila* and *Caenorhabditis*. Analytical strategies for the glycans and glycoconjugates are also summarized.

## 1. Introduction

Glycosylation is a posttranslational modification that ubiquitously occurs in eukaryotes. Compared to higher organisms such as mammals whose glycobiology is being extensively studied, invertebrate glycobiology is frequently neglected and the investigation is limited, fragmentary and unsystematic, partly due to perceived lack of importance compared to that of vertebrates. It is estimated that over 97% of earth's animal species are invertebrates; however, only a limited number of species have been studied with respect to their molecular biology. So far, many invertebrate glycomic studies have focused on recombinant glycoproteins, for example expressed using the baculovirus system. Recent years, increasing numbers of studies have been focusing on the glycomes originally derived from invertebrate species.

*Drosophila* and *Caenorhabditis* are by far the most well-studied invertebrates in glycobiology. As multicellular organisms, they serve as better models compared to lower eukaryotes such as yeast for the investigation of glycosylation and glycoengineering. Some invertebrates can have similar biological activities to those of higher organisms, and yet do not pose safety and ethical issues typically associated with vertebrate models for research purposes. Additionally, the time and cost input for establishing an invertebrate model can be greatly reduced. As a result, the knowledge base for invertebrate glycobiology is continually expanding. Studying invertebrate glycosylation, especially on a model organism, often sheds light on biological functions of the glycoconjugates and assists understanding of glycobiology and targeted glycoengineering in both invertebrates and vertebrates.

This review intends to discuss and summarize the knowledge and research advances related to invertebrate glycosylation, with a focus on the Arthropoda

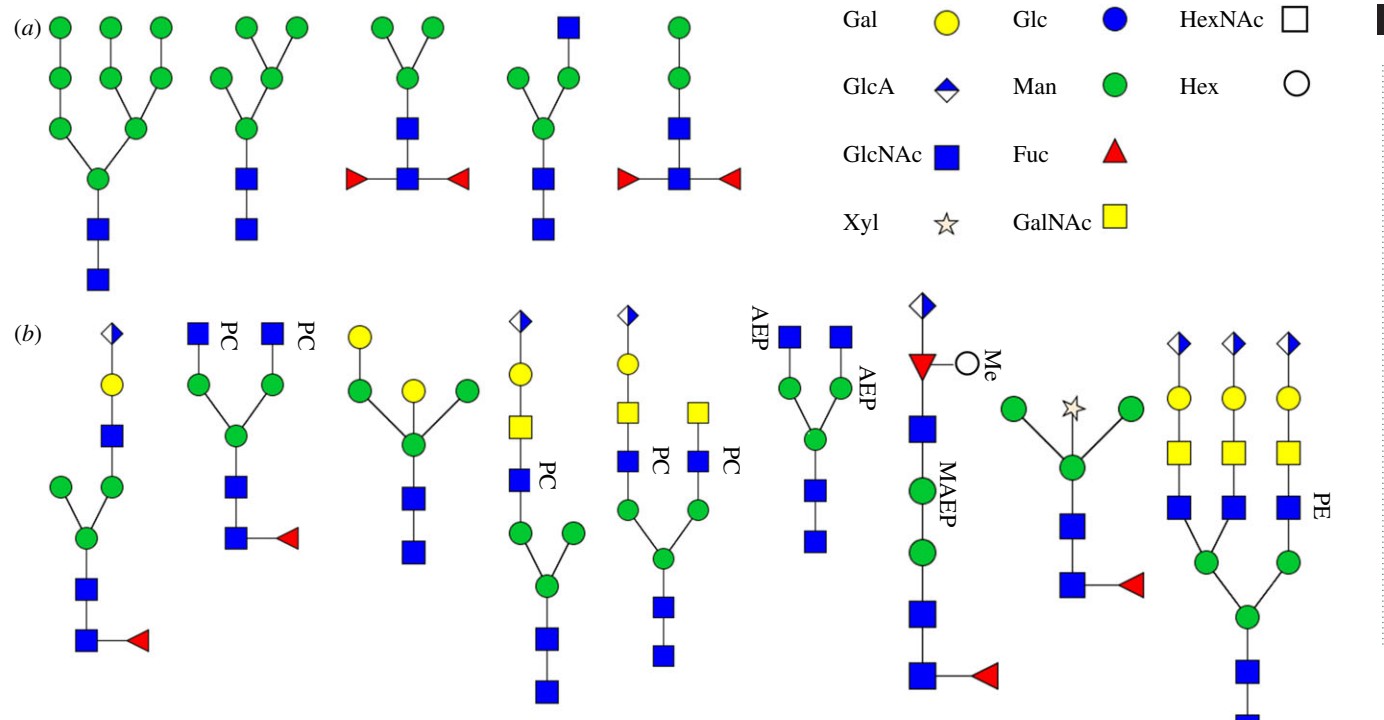

**Figure 1.** (*a*) Examples of *N*-glycans identified in both *Caenorhabditis* and *Drosophila*, whose structures are drawn from the annotated glycan database UnicarbKB. (*b*) Examples of novel *N*-glycans found in invertebrate species, from left to right: *Aedes aegyptii, Pristionchus pacificus, C. elegans, T. ni, L. dispar, Locusta migratoria, V. rubella, Schistosoma mansoni, honeybee royal jelly* [5–11]. Glycans are shown according to the nomenclature of the Consortium for Functional Glycomics. PC, phosphorylcholine; MAEP, methylaminoethylphosphonate; AEP, aminoethyl phosphonate; PE, phosphoethanolamine.

royalsocietypublishing.org/journal/rsob    Open Biol. 9: 180232

model insect *Drosophila* and the nematode worm *Caenorhabditis*. The review aims to focus on the biosynthesis, structures and functions of the glycans and glycoconjugates found in invertebrates, including protein *N*- and *O*-glycans, glycosphingolipids (GSLs) and glycosaminoglycans (GAGs). Meanwhile, analytical strategies for glycan and glycoconjugate analysis will be discussed.

## 2. N-glycosylation in invertebrates

N-glycosylation refers to the attachment of a glycan to the asparagine side chain of a protein. This modification occurs almost exclusively when the asparagine residue is followed by the XT/S sequon regardless of species type, where X refers to any amino acid residue except proline. The highly conserved sequon directs the biosynthesis of *N*-glycans to a protein in the endoplasmic reticulum (ER) and Golgi compartments and ensures that the glycan decoration follows a core machinery. Cells have complex and exquisite machinery for protein glycosylation. Inside animal cells, protein N-glycosylation initiates in the ER, where the carbohydrate moiety $Glc_3Man_9GlcNAc_2$ is synthesized through the action of a series of glycosyltransferases anchored in the ER and then attached to a newly translated protein [1]. The terminal glucoses and mannoses are subsequently removed by their respective glycosidases, and the remaining glycan–protein complex is carried onto the Golgi apparatus for further glycomic editing by different glycosyltransferases and glycosidases, producing variably branched and extended glycan structures [1].

However, the N-glycosylation synthesis routes in invertebrates remain ambiguous and controversial. It is reported that some insects, such as the silkworm *Bombyx mori*, possess a similar glycan synthesis route to that of the mammals [2]. But unlike

vertebrates, the existence of an active *N*-acetylglucosaminidase in the insect cells interrupts the biosynthesis of the complex and hybrid-type *N*-glycans, resulting in almost exclusively high mannose and paucimannose type *N*-glycans [3]. Nevertheless, several species have been found to produce hybrid- and complex-type glycans. *Drosophila*, one of the best studied model insect in glycobiology, possesses similar N-glycosylation machinery to that of vertebrates and produces hybrid- and complex-type *N*-glycans that were once thought absent in insects [4]. However, the relative amount of these complex- and hybrid-type *N*-glycans tends to be low, usually below 10% of the entire *N*-glycan pool. Another well-studied invertebrate, the nematode *Caenorhabditis*, also contains a nearly contiguous series of *N*-glycans (figure 1) [12]. *Caenorhabditis* was also found to contain fuco-paucimannosidic and bisecting fucose–galactose branched glycans that are unique to this nematode [5,6,12].

Recently, dipteran species, specifically mosquito larvae, were found to contain sulfated and glucuronylated antennae [13], indicating that insect glycans can have higher levels of structure complexity and variation than previously expected. In addition to sulfated and glucuronylated, core difucosylated and zwitterion phosphorylcholine and phosphoethanolamine-modified *N*-glycans were also identified in a handful of invertebrate species such as *Trichoplusia ni* and *Lymantria dispar* (figure 1) [7]. The mollusc *Volvarina rubella* was also found to contain novel *N*-glycans with phosphonate and phosphorylcholine modifications in addition to Fuc and GlcA modifications (figure 1) [8,9]. Additionally, xylosylated glycans and triantennary phosphoethanolamine-modified glucuronylated glycans have also been identified from *Schistosoma* and honeybee royal jelly, respectively (figure 1) [10,11]. These findings have vastly expanded the current insect glycan repertoire and enabled a fresh look at invertebrate glycan structures and their functions.

royalsocietypublishing.org/journal/rsob    Open Biol. 9: 180232

The majority of membrane and secreted proteins are cotranslationally N-glycosylated and are involved in a broad range of biological activities. *N*-Glycoproteins are found to be involved predominantly in cell–cell adhesion, body growth, embryonic development and organ development [14]. Cell glycan biosynthesis is facilitated by a few hundred enzymes, including glycosyltransferases, glycosidases and enzymes related to sugar modification, metabolism and transport, etc. [15]. The glycoenzyme set involved in protein N-glycosylation appears to be different between different species orders, as evidenced by highly conserved N-glycoproteomes within their respective phyla and different *N*-glycan antennal modifications between evolutionarily distant species [14,16]. Extensive studies have shown that mutations of any of the glycoenzymes are likely to cause serious morphological and developmental defect or even death in many invertebrate species. The *Drosophila alg5* gene, which codes for an enzyme involved in the early steps of protein N-glycosylation, is essential for the correct epidermal differentiation during *Drosophila* late embryogenesis [17]. Mutation of the *Drosophila* gene *xit*, which encodes an enzyme involved in the addition of the terminal glucose to the *N*-glycan precursor, impairs cell intercalation in the lateral epidermis during germband extension and apical constriction of mesoderm precursor cells [18]. Thus, identification of invertebrate genes encoding the glycoenzymes involved in glycan biosynthesis seems highly necessary in order to facilitate deeper understanding of invertebrate glycosylation and to precisely engineer desired glycan structures using invertebrate models. In fact, genetically engineered mutants of *Caenorhabditis* and *Drosophila* have been established in order to reveal the biological functions of specific glycoenzymes such as fucosyltransferases [5], *N*-acetylglucosaminyltransferases [19], hexosaminidases [20,21] and other glycoenzymes [22,23].

# 3. Profiling of glycans and glycoconjugates in invertebrates

The structural diversity and heterogeneity of glycoconjugates is attributed mainly to the attached glycans. Unlike proteins and nucleotides, glycan structures can be highly branched, and the monosaccharide units composing the glycans are often isomeric, leading to increased structural complexity. Additional structural complexity arises from the different glycosidic linkages between the monosaccharides as well as further decoration of the monosaccharides by sulfation and phosphoethanolamine. Bacteria and archaea have the broadest diversity of monosaccharides, with a total of approximately a hundred different types of monosaccharides [24]. Plants are the next, and then eukaryotic animals. There are 11 monosaccharides typically found in invertebrate glycans: glucose, mannose, galactose, *N*-acetylglucosamine (GlcNAc), *N*-acetylgalactosamine (GalNAc), xylose, fucose, *N*-acetylneuraminic acid (NeuAc), *N*-glycoylneuraminic acid (NeuGc), glucuronic acid (GlcA) and iduronic acid. Further modifications of the monosaccharides such as methylation, sulfation and zwitterionic modification including phosphorylcholine, phosphoethanolamine and aminoethyl phosphate have also been identified in invertebrates.

Because of the high structural complexity, structural analysis of glycans and glycoconjugates remains very challenging. The main workhorse in today's glycomic and glycoproteomic analysis is the mass spectrometry (MS)-based approach due

to its high-sensitivity and high-throughput capabilities. Auxiliary methods, such as glycosidase digestion, and less frequently nuclear magnetic resonance spectroscopy, could also be used as complementary tools to confirm carbohydrate structures. In MS-based approaches, improving glycan detection sensitivity and assigning MS spectra are among the most challenging steps. Here, in this part, we will briefly discuss large-scale mapping of glycans and glycoconjugates in invertebrates by MS-based techniques.

Currently, large-scale identification of N-glycosylation sites have been quite routine. More than a thousand insect proteins have been found to be N-glycosylated. Zielinska *et al.* [25] developed a filter-aided deamidation method which takes advantage of the enzymatic deglycosylation reaction that turns the asparagine residue into aspartic acid. When the reaction is carried out in $^{18}$O water, a fixed mass shift of 2.989 occurs, which can be readily detected by mass spectrometry and analysed automatically using software platforms such as MaxQuant [26]. Using this method, the N-glycoproteomes across seven evolutionarily distant species were mapped, including *Arabidopsis thaliana*, *Schizosaccharomyces pombe*, *Saccharomyces cerevisiae*, *Caenorhabditis elegans*, *Drosophila melanogaster* and *Danio rerio* [14]. *N*-Glycoproteins are found to be almost exclusively located outside the cell including the cell membrane, or in anticipated intracellular organelles such as ER and Golgi [14]. Analyses of N-Glycoproteome orthologues within and across different phyla indicate that each phylum has characteristic N-glycoproteomes that are distinct from species in other phyla [14]. Therefore, it seems necessary, not just out of curiosity, to investigate the N-glycoproteomes across different species, especially in insects that currently have a limited N-glycoproteome revealed, in order to fully understand the functions of protein glycosylation.

Although the filter-aided deamidation method can perform large-scale identification of protein glycosylation sites, it cannot provide information on the specific glycan structures attached to the glycosylation site. Typically, there are two means to obtain glycan structural information. The first way is to isolate the glycans from proteins and analyse the glycome separately. Currently, automated and high-throughput glycomic profiling is under rapid development and has achieved great progress. In typical high-throughput glycomic analysis, glycans are cleaved from proteins either by PNGase or by other chemical methods, enriched and then labelled before LC–MS analysis [27]. Glycan structures can be deciphered by software such as MultiGlycan [28], SimGlycan [29] and GlycoWorkBench [30], which has greatly promoted large-scale structural analysis of glycans with various labelling techniques. A handful of invertebrate species, such as *Drosophila* [31,32] and *Caenorhabditis* [33,34] have had their glycome analysed, which has significantly renewed the understanding of structural and functional glycobiology in invertebrates. That said, in-depth analysis of the N-glycomes, especially finding novel glycan structures, can still be quite challenging and far from routine. A combination of exoglycosidase digestion, offline LC separation and purification, as well as MALDI-TOF MS/MS or LC/MS/MS analysis is usually needed in order to reveal new glycan structures [19].

The other way to obtain glycan structural information is to study the glycopeptides with the glycans attached, which is even more challenging compared to glycomic study alone, because the former involves peptides and glycan identification

simultaneously. Developing robust and reliable pipelines for large-scale profiling of glycopeptides is still under way. The most challenging steps during the pipeline development, other than developing high-sensitivity MS detection methods for glycopeptides, perhaps is automated assignment of MS and MS/MS spectra to specific glycopeptides. Currently, software such as GLYCOMASTER DB [35], BYONIC [36] and ARMONE [37] has been developed to perform automated large-scale analysis on intact glycopeptides based on MS fragmentation datasets. So far, the glycopeptide identification pipelines have been quite successfully applied in studying large glycoproteomes for mammalian tissues and organs; however, very few reports have been focused on large-scale profiling of invertebrate glycoproteomes.

## 4. O-glycosylation in invertebrates

Depending on the first monosaccharides linked to the protein, O-glycosylation can be further divided into O-GalNAcylation, O-mannosylation, O-fucosylation and so on. Current knowledge on invertebrate O-glycosylation is still very rudimentary. In contrast with N-glycosylation, there is no consensus sequence for O-glycosylation. Serine and threonine are the most common accepting amino acids for these modifications. Tyrosine, hydroxylysine and hydroxyproline are also possible residues for O-glycosylation. In addition, O-glycan biosynthesis is carried out by the addition of monosaccharide one after another, which is different from the case of N-glycosylation. O-GalNAcylation is one of the most common forms of O-glycosylation, and was first identified in mucin, a heavily glycosylated protein 40% of whose molecular weight is occupied by glycans. Therefore, O-GalNAcylated glycans are also called mucin-type glycans. Approximately 90% of all O-glycans in Drosophila belong to the mucin type [38]. In mucin-type O-glycosylation, a GalNAc residue is directly linked to serine or threonine for the initiation of protein glycosylation. In invertebrates, mucin-type glycans are found extensively in secreted proteins or the extracellular part of membrane proteins. Currently, except mucin-type O-glycans, other types of O-glycans in invertebrates have not been investigated in detail. There are eight common O-glycosylation cores found in mammals [39]; however, few of these have been found to be present in invertebrates. Examples of mucin-type O-glycans for the model organisms Caenorhabditis and Drosophila are shown in figure 2.

In invertebrates, various UDP-GalNAc transferases initiate O-GalNAcylation by modifying the Ser/Thr residues with a GalNAc [40]. Following extension of the glycan chain results in several different core structures. For the core-1 structure, β1−3 galactosyltransferase adds an additional galactose to the GalNAc residue. Additional glucuronic acid or GlcNAc residues can be found to further modify the glycan chain (figure 2). Core-1 type and extended core-1 mucin-type glycan structures have been identified in Caenorhabditis and Drosophila (figure 2). Caenorhabditis also synthesizes core 1 mucin-type glycans substituted on Gal and/or GalNAc by Glc residues, similar to those in vertebrates [41]. So far, sulfated O-glycans have been identified in Drosophila but not Caenorhabditis [5], and phosphoethanolamine modification has been reported in the insect Vespula germanica [42], suggesting the potential of broad-spectrum modifications to invertebrate O-glycosylation that are waiting to be discovered.

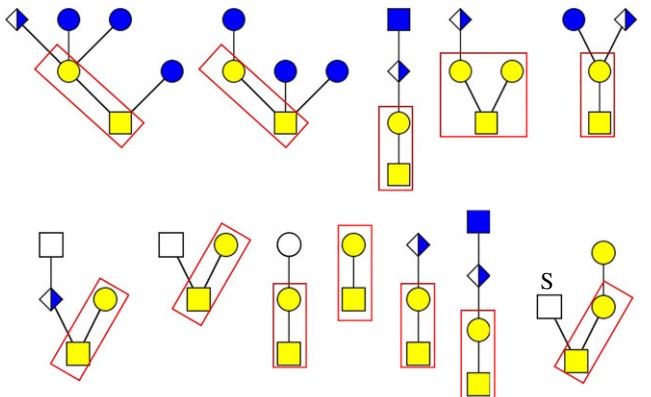

**Figure 2.** Examples of mucin-type O-glycans in Caenorhabditis (upper row, from annotated glycan database UnicarbKB) and Drosophila (lower row) [38]. Red rectangles denote common O-glycan cores.

Mucin-type O-glycan modification is critical for the development and function of multicellular organisms. In Drosophila, the pgant gene family, which is responsible for encoding the GalNAc transferases, is essential for viability of the insect. O-GalNAcylation regulates essential developmental programmes and modulates trafficking through the secretory pathway, and mutation or silencing of glycoenzymes such as the pgant genes results in malfunctions in cell adhesion and Golgi trafficking [43]. However, O-glycosylation functions in other invertebrate species have not been fully elucidated and need further investigation.

O-fucosylation and O-glucosylation are typical modifications to protein epidermal growth factor (EGF)-like domains and thrombospondin type 1 (TSP1) domains. The O-fucosylation site of the EGF domain is usually flanked by cysteine residues that form disulfide bridges, which can be recognized by O-fucosyltransferase (OFUT)1, whereas in TSP1 domains another O-fucosyltransferase, OFUT2, recognizes and directs the transfer of fucose to the protein. In Drosophila, mutations to OFUT1 lead to multiple organ defects and lethality [44,45]. The addition of a GlcNAc residue to the fucose is accomplished by Fringe, an N-acetylglucosaminyltransferase in Drosophila that is involved in many functions such as eye development [46] and adult eclosion and survival [47]. In O-glucosylation, transfer of glucose to the serine/threonine residue is achieved by a single glucosyltransferase named Rumi [48], and the addition of xylose to the glucose is accomplished by the enzyme Shams [49]. In Drosophila protein Notch, 22 sites were found to be O-fucosylated and 18 were O-glucosylated [50]. O-glucosylation and O-fucosylation function cooperatively and play important roles in Notch transport and signalling in Drosophila [51].

O-Mannose glycans constitute less than 1% of the total Drosophila glycan pool [38]. O-mannosylation has been gaining interest due to its conserved process across most eukaryotes and defects in this modification give rise to human diseases [52]. The transfer of mannose to serine or threonine is accomplished by the enzyme family O-mannosyltransferases. In Drosophila, O-mannosylation occurs in the protein known as dystroglycan. The Drosophila homologues of two O-mannosyltransferase genes, POMT1 and POMT2, function in association with each other to maintain normal muscle development [53]. Another type of O-mannosylation, such as for the cadherin superfamily, depends on the TMTC-type mannosyltransferases for O-mannosylation [54];

however, their mannosyltransferase activity in *Caenorhabditis* and *Drosophila* is yet to be proven. In *Drosophila*, POMT mutation results in rotated abdomen, defective synaptic transmission and muscle dystrophy [53,55]. In humans, mutations to these two transferases cause autosomal recessive disorder, which leads to malfunction in the brain, muscle and eye [56]. A summary of the main functions of invertebrate glycans is presented in table 1.

Compared to *N*-glycan analysis, *O*-glycan poses more analytical challenges, partly because there are currently no universal enzymes available to cleave off *O*-glycans from the proteins. General strategies to removed *O*-glycans from proteins are through chemical methods such as alkaline β-elimination or hydrazine hydrolysis. In addition, O-linked glycans do not have a common core structure, and instead can have more than eight types of core structures. Furthermore, unlike *N*-glycans which appear to be synthesized following predefined structural antennae, *O*-glycans seem to branch more irregularly, making it difficult to define the specific structure. Often times, glycosidases are used in addition to mass spectrometry to determine the definite structures.

## 5. *O*-GlcNAc modification

O-linked GlcNAc modification refers to the addition of a single GlcNAc residue to the serine or threonine residue of a protein. This modification is reversible and highly dynamic in that GlcNAc is added and removed regularly depending on the cellular environment by two unique enzymes, the O-linked GlcNAc transferase (OGT) and the O-linked GlcNAcase (OGA). In contrast with other types of O-glycosylation, O-GlcNAcylation can actually occur in the nucleus and cytosol rather than in the ER and Golgi apparatus; therefore, O-GlcNAcylation is in nature more akin to protein phosphorylation than typical O-glycosylation. This modification is particularly heavily present on proteins involved in signalling, stress response and energy metabolism such as nuclear pore proteins, phosphatases, metabolic enzymes, etc. [68]. O-GlcNAcylation regulates protein translation, stability and turnover, and has been demonstrated to be engaged in neurodegenerative diseases, diabetes and cancer [57–59]. Alteration of *O*-GlcNAc profiles of several proteins in the pancreatic β-cell has been reported to associate with the upregulation of insulin secretion from the pancreas [60]. There is also an extracellular form of O-GlcNAcylation on EGF repeats mediated by the EOGT enzyme in the ER [69].

O-GlcNAc modification is ubiquitous and essential in multicellular organisms. Mutations of genes relating to the GlcNAc modification will cause severe growth phenotypes or even death. The deletion of OGT or OGA from *C. elegans* results in up- and down-regulation of hundreds of transcripts, which is likely due to the misguided *O*-GlcNAc modification of RNA polymerase II as well as the basal transcription complex, indicating the role of O-GlcNAcylation in regulating transcription [70]. The disruption of the OGT gene in *C. elegans* has been shown to induce metabolic disorder and reduced lifespan [71]. A similar metabolic disorder was also observed in the species with OGA gene disruption, but interestingly its lifespan was extended. In *Drosophila*, the extent of protein O-GlcNAcylation was found to increase with the developmental stage [72]. Shortened lifspan was observed in an OGT gene

**Table 1.** Invertebrate glycosylation types and functions.

| type | exemplary structure | main known functions | references |
|---|---|---|---|
| *N*-glycan | | cell–cell adhesion, body growth, embryonic development and organ development | [14,17,18] |
| *O*-glycan | mucin-type *O*-glycan | cell adhesion, Golgi trafficking and organ development | [38,43] |
| | *O*-fucose glycan | Notch transport and signalling | [44–47,49–51] |
| | *O*-glucose glycan | | |
| | *O*-mannose glycan | growth and embryonic/organ development | [53,55,56] |
| | *O*-GlcNAc glycan | regulating transcription, protein functions and metabolism | [57–60] |
| glycosphingolipid glycan | | host–pathogen interactions, cell recognition, development and modulating transmembrane signalling | [61–63] |
| glycosaminoglycan | | host–pathogen interactions, embryogenesis, regenerative and developmental roles | [64–67] |

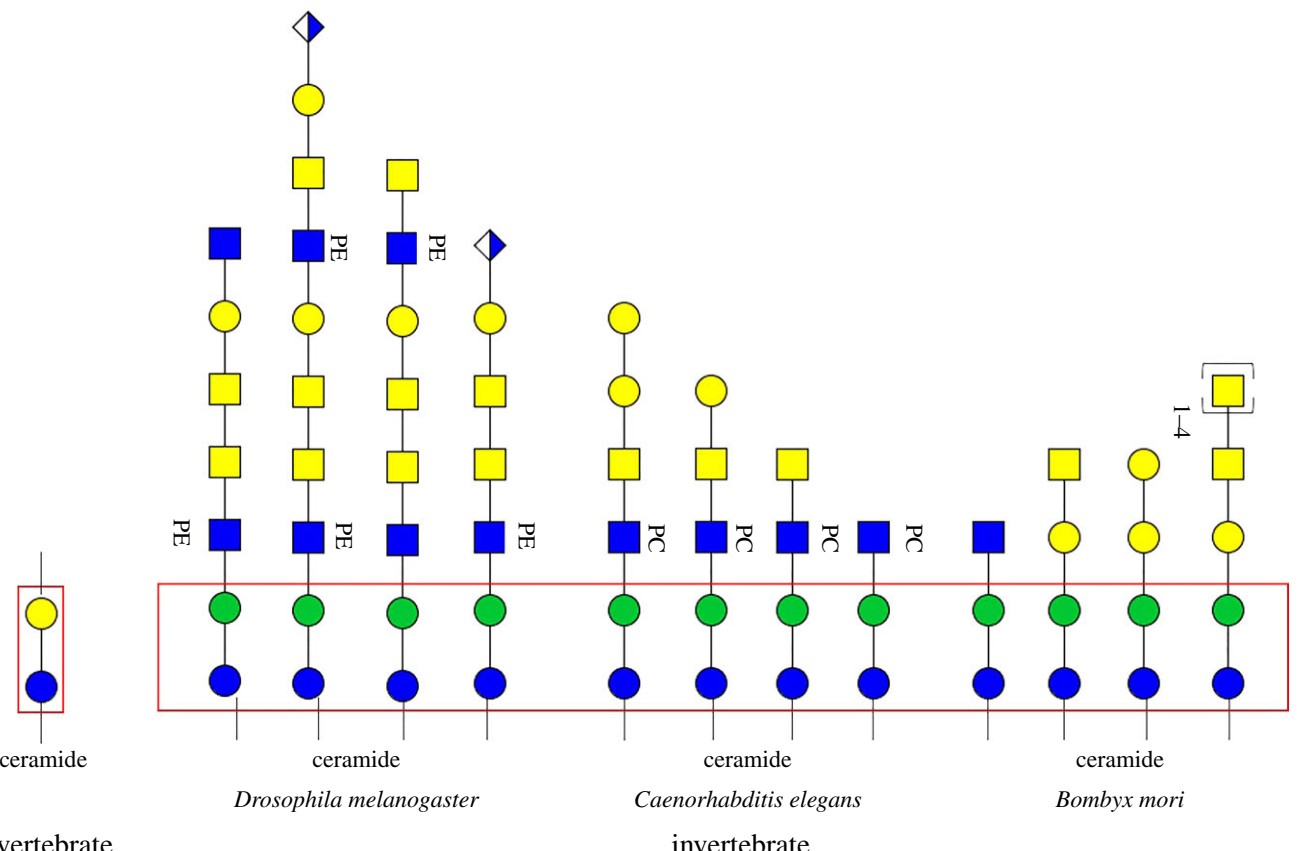

**Figure 3.** Exemplary GSL structures in invertebrate species *D. melanogaster* [82,85], *C. elegans* [86] and *B. mori* [87]. Red rectangles denote GSL cores. PC, phosphorylcholine; PE, phosphoethanolamine.

mutant of *Drosophila*, which survived through the larval stages but died in the pupal stages [73]. Because the OGT gene is highly conserved throughout species, the OGT gene in humans has been transgenically introduced for the rescue of OGT null *Drosophila*.

Most current studies on O-GlcNAcylation have been focused on the functional aspects; profiling of *O*-GlcNAc modified proteins has been reported in mammals but seldom in invertebrates. As a structurally simple modification, *O*-GlcNAc modification is even more difficult to detect and determine. First, as mentioned above, *O*-GlcNAc modification is dynamically cycling on and off the modified peptide depending on the cell environment, which is easily deglycosylated by the glycosidases that exist in the cell during cell lysis. To prevent autodeglycosylation, an *O*-GlcNAc glycosidase inhibitor, such as PUGNAc, is added during cell lysis and sample preparation. Another issue is associated with O-GlcNAcylation detection, such as by mass spectrometry, the most popular technique currently used for *O*-GlcNAc analysis. The attached *O*-GlcNAc is sensitive to the electrospray ionization process and readily falls off from the peptide backbone. Last but not least is the lack of efficient and widely applicable enrichment methods for O-GlcNAcylated peptides prior to MS detection. The traditional glycan enrichment method lectin affinity is not quite applicable in the case of O-GlcNAcylation, because when only a single GlcNAc residue is attached to the peptide, the interaction between O-GlcNAcylated peptide and the lectin is very weak. Nevertheless, by taking advantage of this weak interaction, Vosseller *et al.* [74] developed a lectin weak affinity chromatography method for targeted enrichment of O-GlcNAcylated peptides. Using a similar method, more than 1700 O-GlcNAcylated peptides have been identified in mouse

synaptic membrane [75]. O-GlcNAcylated peptides can also be enriched via immunoprecipitation using antibodies specific to O-GlcNAc, such as RL2 [76], CTD110.6 [77], 18B10.C7 [78], etc. Click chemistry-based method has also become quite popular in recent years for the enrichment of O-GlcNAcylated peptides [79]. In this method, *O*-GlcNAc residues are either enzymatically grafted with an azide-tagged GalNAz or metabolically incorporated with an azide-tagged GlcNAc residue, and labelled with biotin and subsequently enriched by avidin or streptavidin immobilized solid matrix [80].

## 6. Glycosphingolipids in invertebrates

GSL is a type of glycolipid commonly found in animals. In GSLs, glycans are covalently linked to a ceramide lipid moiety that is composed of a long-chain alcohol known as sphingosine in amide linkage to a fatty acid. GSLs are structurally diverse mainly due to variable sugar modifications to the ceramide, and the glycan moiety of invertebrate GSLs differs notably from those of vertebrates. Vertebrates have a GSL core disaccharide Gal(β1–4)Glcβ linked to the ceramide, whereas for invertebrates the most common core disaccharide is Man(β1–4)Glc [81–83], except for the mollusc *Aplysia kurodai*, whose GSL core is the same as in vertebrates [84]. In addition, the ceramide moiety of invertebrate GSLs is found to be different from those of vertebrates, in that the sphingosine chain is shorter for invertebrates [81].

Further extension of the glycan core structure is species-specific. The highly diversified and individually tailored GSL structures indicate their important roles in developmental or tissue-restricted functions. In *Drosophila*, the core

mactosyl Man(β1−4)Glc structure can be further modified with a GalNAc(β1−4)GlcNAc(β1−3) residue, and the terminal Gal can be further capped with GlcA (figure 3) [82,85]. Phosphoethanolamine is present as a typical modification to dipteran glycolipids and aminoethylphosphonate to those of molluscs [8,13,84]. In *Caenorhabtidis*, the GSLs were reported to consist of the core structure GlcNAc(β1−3)Man(β1−4)Glc(β1)Cer, similar to that in arthropods (figure 3) [86]. Also, phosphorylcholine is a known antenna component of nematode glycolipids (figure 3) [88,89]. Recently, GSLs of the lepidopteran species *B. mori* were investigated, which contain the same conserved core structure, but novel extensions were revealed (figure 3) [87]. The major GSL components for *Manduca sexta* were identified as mactosyl ceramide [81]. Gangliosides, GSLs modified with sialic acid residues, have not been reported for invertebrates to date.

In invertebrates, GSLs play important roles in host–pathogen interactions, cell recognition and body development [61,90]. Elimination of the *egh* and *brn* genes, which encode mannosyltransferases and N-acetylglucosaminyltransferases that are responsible for the early-step biosynthesis of GSLs, is lethal to *Drosophila* [62]. Mutation of either of the two genes caused overproliferation of neural cells and enlarged peripheral nerves, phenotypes similar to human neurofibromatosis diseases [63]. The results indicate that, like vertebrates, GSLs play pivotal roles in cell recognition and modulating transmembrane signalling.

Although invertebrate GSLs are no longer thought to be GSL-free, profiling of GSL structures have been limited to only a few invertebrate species and the molecular details associated with GSLs are largely unknown. Glycolipid profiling is gaining growing interest motivated by the important biological roles of the glycan head groups. The analysis of GSLs involves determination of both the ceramide and glycan moieties, both of which have high structural diversity. In general, glycolipids are extracted from tissues or body fluids by chloroform–methanol extraction. Separation and analysis of GSLs relies largely on a combination of techniques, such as thin-layer chromatography [87], gas chromatography [62], nuclear magnetic resonance [62,87] mass spectrometry [91], etc., and mostly count on manual annotation and interpretation of the GSL data. High-throughput workflows are still under development. A database and software for the MS analysis of ganglioside and sulfate-modified GSLs are under development for automated interpretation of MS data of the GSLs [92].

# 7. Glycosaminoglycans in invertebrates

GAG is a linear polysaccharide consisting of repeating disaccharide units covalently linked to a protein (proteoglycan). The most abundant cell surface GAG structural subtypes include heparan sulfate characterized by disaccharide unit GlcA(β1−4)GlcNAc(α1−4) and chondroitin sulfate characterized by GlcA(β1−3)GalNAc(β1−4). Variable degrees of sulfation and GlcA epimerization (to iduronic acid) may occur on the GAG backbone. While vertebrates tend to have additional GAG types such as dermatan sulfate and hyaluronic acid, most invertebrates were reported to contain only heparan sulfate and chondroitin sulfate with or without sulfate.

More than 20 chondroitin sulfate proteoglycans have been found in the model organism *Caenorhabditis*, indicating previously underestimated GAG structural and functional heterogeneity in invertebrates [33]. Chondroitin chains in *Caenorhabditis* were once thought not sulfated due to the lack of relative sulfotransferases and epimerase for modifying the sugar residues [93,94]; however, a recent study found an active chondroitin sulfotransferase in this species and its sulfated chondroitin chains were revealed, albeit at a low level [95]. In addition, the *Caenorhabditis* proteoglycan core proteins were found to be different from those found in vertebrates or the invertebrate *Drosophila* [93,96]. The common tetrasaccharide core linking the repeating disaccharides and the serine residue of the proteoglycan is reported as GlcA(β1−3)Gal(β1−3)Gal(β1−4)Xyl for chondroitin and heparan [97], whereby the relevant enzymes synthesizing the core are encoded by genes defective in *sqv* mutants [98]. Recently, a novel GAG tetrasaccharide core with additional galactose and phosphorylcholine modifications was reported for the parasitic nematode *Oesophagostomum dentatum* [99].

Degrees of sulfation on the GAG disaccharide backbone are highly conserved within a class but significantly different between classes of invertebrates [94], indicating evolutionarily distinct functionalities. For example, *Drosophila* chondroitin sulfate is composed of 71% HexA-GalNAc and 29% HexA-GalNAc (4-*O*-sulfate) [100], whereas that of the Chelicerata species *Trachypleus tridentatus* is composed of 46% HexA-GalNAc (4-*O*-sulfate) and 54% GlcA(3-*O*-sulfate)-GalNAc(4-*O*-sulfate) [101]. GAGs are present covalently linked to proteins via type-specific linkages forming proteoglycans. As for *Caenorhabditis*, the GAGs in *Drosophila* are based on the same canonical GlcA-Gal-Gal-Xyl core for attachment to proteins; some of the relevant enzymes have been characterized, such as oxt [102] and GalT7 [103].

GAGs are ubiquitously found on the surface and extracellular matrix of mammalian cells, interacting with various ligands and playing crucial roles in many pathophysiological processes. The chondroitin sulfate GAGs are structural constituents of complex matrices such as cartilage, brain, intervertebral discs, tendons and corneas. Genetic studies on the model organism *Drosophila* showed that heparan sulfate GAGs act as core receptors for many growth factors, and participate in the generation and long-range maintenance of gradients for morphogens during embryogenesis and regenerative processes [64]. Knockdown of GAGs in *Drosophila* reduces the binding of α C protein, a virulence determinant of group B streptococcus, resulting in longer host survival [65]. The results indicate that host cell surface GAGs are vital during pathogen invasion and that interfering with the binding of this sugar may protect the host against infection. Mutation of a heparin sulfate proteoglycan homologue in *Drosophila* leads to cell cycle arrest of neuroblasts in the larval brain [66,67]. The *Drosophila ttv* gene, which encodes an enzyme responsible for adding monosaccharide to the GAG backbone, is a homologue of the mammalian Ext class of tumour suppressor genes that cause human bone dysplasia [104,105].

Typical GAG and proteoglycan analyses use bottom-up approaches. The proteoglycans are extracted by strong denaturing agents and purified by ion exchange or size exclusion chromatography. The GAG moiety can be isolated from the protein via β-elimination or hydrazinolysis, and its disaccharide unit can be degraded by bacterial lyases such as heparinase or chondroitinase, and further derivatized before analysis by gas chromatography, liquid chromatography and/or mass spectrometry [106–108]. However, only compositional information of the disaccharide building blocks can be derived

royalsocietypublishing.org/journal/rsob Open Biol. 9: 180232

from the bottom-up approach, and no sequence information can be obtained. Top-down mass spectrometry analysis of GAGs is possible for the direct structure and site determination of intact proteoglycan and GAG structures, but may require pure proteoglycans, which is rather difficult [109]. Additional difficulties arise from the assignment of the GAG MS/MS spectrum for top-down analysis due to the variation of sulfation sites between the disaccharide units. Software programs are being developed for automated annotation of the GAG fragmentation spectra to assist structure elucidation of GAG compositions and sequences [110].

# 8. Concluding remarks

Glycosylation confers heterogeneity on glycoconjugates and finely tunes their structures and functions [111,112]. In vertebrates, aberrant glycosylation can be indicative of various disease states [113,114]. In invertebrates, disturbance of glycosylation pathways has been demonstrated to cause serious defects, such as abnormal metamorphosis and even mortality. The investigation of invertebrate glycosylation often provides new insight into mechanisms underlying physical/neurological impairments in vertebrates and helps to establish novel therapeutic treatment strategies.

It is also found that invertebrate glycomic profiles can change upon alteration of physiological, pathological or developmental stages. A mutant *Caenorhabditis* that is resistant to bacterial infection is observed to be deficient in many *N*- and *O*-glycans compared with its wild-type [115]. Similarly, *Caenorhabditis bre-1* mutant, which encodes an enzyme that catalyses biosynthesis of GDP-mannose to GDP fucose, was found deficient in fucosylated glycoconjugates and resistant to the toxin *Bacillus thuringiensis* [22]. Our laboratory also found that the N-glycomic profiles of *B. mori* alter after BmNPV viral infection (F Zhu, D Li, D Song, P Lv, Yao Q, Chen K 2019, unpublished data). In addition, the monosaccharide profiles of *B. mori* nervous system change with different development stages [116]. Therefore, it seems necessary, not just out of curiosity, to profile the glycoproteomes across different development stages of species, which will contribute to deeper understanding of the functional roles of glycosylation.

Data accessibility. This article has no additional data.

Authors' contributions. F.Z. wrote the manuscript, D.L. formatted and edited the manuscript and K.C. revised the manuscript. All authors gave final approval for publication.

Competing interests. We declare we have no competing interests.

Funding. Work in the authors' laboratory has been funded by the National Natural Science Foundation of China (31702186 and 31872425), Natural Science Foundation of Jiangsu Province China (BK20160509) and China Postdoctoral Science Foundation (2016M601725).

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
