## [Reviewer comments · Open Biology]

Review History

RSOB-18-0232.R0 (Original submission)

Review form: Reviewer 1

Recommendation

Accept with minor revision (please list in comments)

Are each of the following suitable for general readers?

- a) **Title**
Yes
- b) **Summary**
Yes
- c) **Introduction**
Yes

Is the length of the paper justified?

Yes

Should the paper be seen by a specialist statistical reviewer?

No

Is it clear how to make all supporting data available?

Not Applicable

Is the supplementary material necessary; and if so is it adequate and clear?

Not Applicable

Do you have any ethical concerns with this paper?

No

Comments to the Author

As stated by the authors, the rich diversity of invertebrate species, genomes, and the time and cost efficiency of raising and experimenting on these species have enabled a handful of the species to become excellent model organisms.

This is an interesting review and the background, the objectives of the work have been properly described. Before accept, there are still some points that I want to discuss with the authors to make the review be clearly addressed as below.

1. Page 2, line 19-22, "Studying invertebrate glycosylation, especially... assists understanding glycobiology and targeted glycoengineering in both invertebrates and vertebrates." What is the most differences between the invertebrates and vertebrates in the glycobiology and targeted glycoengineering? Please add the most concise summary to this part.
2. Page 4, Figure 1, Examples of N-glycans in invertebrate sepecies, I can not find the relative description of this fiugre in the text. Please position it.
3. Page 13, Table 1, also I can not find the description in the text. Please position it in the text.
4. Page 21-22, Concluding remarks The conclusion is so long which will make the paper not clear and comprehensive, so please refine the conclusion carefully.

Review form: Reviewer 2

Recommendation

Major revision is needed (please make suggestions in comments)

Are each of the following suitable for general readers?

- a) **Title**
Yes
- b) **Summary**
Yes
- c) **Introduction**
Yes

Is the length of the paper justified?

Yes

Should the paper be seen by a specialist statistical reviewer?

No

Is it clear how to make all supporting data available?

Not Applicable

Is the supplementary material necessary; and if so is it adequate and clear?

Not Applicable

Do you have any ethical concerns with this paper?

No

Comments to the Author

Open Biology, Zhu et al.

The authors have sought to summarise data on glycans in invertebrates, especially *Drosophila* and *Caenorhabditis*. However, there are some inaccuracies in this review which must be corrected regardless of the final publication decision for Open Biology - I will not claim to have found all mistakes and grammatical errors. I also do not seek that it be comprehensive, but it must be accurate.

Page 1, Line 5: The first sentence of the abstract is one of the most common statements about glycosylation - the exact same phrase can be found in Ceciliani et al, 2007, in Hayes et al, 2012 or Cornelissen et al, 2016 (to name the first few google 'hits'). The authors should seek to have at least the first sentence different to those elsewhere in the literature.

Line 18: 'on the model organisms' (plural), then spelling of *Caenorhabditis* (not *Carno* ...)

Page 2, Line 6: "97%" would be of animal species not of all the earth's species - put a semi-colon (;) after the word 'invertebrates'.

Line 8: I would not say that the 'vast majority' of invertebrate glycomic studies have been on recombinant proteins - certainly a large number, but what about individual glycoprotein antigens from *Echinococcus*, glycoprotein allergens from insect venoms or haemocyanins from snails, never mind the various whole-organism studies?

Page 3, line 15 - not 'inside the body of mammals', rather 'inside animal cells'.

line 18: 'terminal' rather than 'end'

line 21: 'glycoSYLtransferases'

Figure 1: panel A is ok, but for panel B - *Trichomonas* is not an invertebrate species (note spelling of species) - it is a protist; the authors could put a PC on the structure in *Lymantria* (PC also found in *Trichoplusia ni*), add a trifucosylated core for *C. elegans* or a GalFuc core with methylaminoethylphosphonate from a marine snail, a xylosylated glycan from some species (e.g., *Schistosoma* or snail) or triantennary PE-modified glucuronylated glycans from royal jelly (recent paper in MCP). They should also add mannose to the set of monosaccharides. AEP is aminoethylphosphONate.

Page 4, line 13 - write 'insect cells' not 'insect body'.

Page 5, line 8 - what is meant by a high degree of structure conservation with those in mammals? which structures? Of course, three antennae are possible in *C. elegans*, but these are not galactosylated - the galactose residues being on fucose residues or on mannose (e.g., bisecting position).

line 11 - write 'dipteran species, specifically mosquito larvae, were ...'

line 15 - PE is phosphoethanolamine; Volvarina (Eckmair, MCP, 2016) is a mollusc and does not fit so well in the same sentence as two moths - but could be mentioned elsewhere - with branched Fuc, GlcA modifications, phosphonate and a trace of PC.

line 23 - 'glycosyltransferases'

Page 6, line 15 - a number of glycomics studies have been done on gene knock-out strains in *C. elegans* - e.g., fut-1/fut-6/fut-8, gly-12/gly-13/gly-14, hex-2/hex-3, aman-2, bre-1. RNAi has also been done on glycogenes from *C. elegans*. Also the nac mutant (GDP-Fuc) transporter in *Drosophila* was investigated glycomically (Geisler, JBC).

Page 7, line 4 - ribose is not found in invertebrate N-glycans, but GalNAc is; check spelling of N-glycoylneuraminic. Further variation with methyl groups or anionic/zwitterionic groups could be mentioned.

Page 7, line 16 and pages 8/9 - the routine nature might be 'large scale N-glycoproteomics' to identify glycosylation sites, but the in-depth analysis to find new structures in invertebrates is far from routine and certainly Glycoworkbench is not ok for dealing with novel structures! The 'analytical' discussion focuses on glycoproteomics and does not mention the great advances recently made with off-line LC-MALDI-MS in order to find new structures.

Page 9, line 16 - suggest 'O-glycosylation in invertebrates' (also without word 'the')

Page 10, line 11 - 'however, few of these have been ...'

Figure 2: the first structure shown might well be a breakdown product of a GAG; structures from *Drosophila* with GlcA could also be drawn, based on the literature - far from everything is in UnicarbKB!

Page 11, line 5 - actually if looking at Kurz et al, Figure 13 and Suppl Figure 11, then it is shown that *Drosophila* has sulphated O-glycans; *Vesputula* has PE (not PC) on its O-glycans.

line 13 - spelling of Golgi (not 'Gogi')

line 23 - fringe is not a 'glucotransferase' but an 'N-acetylglucosaminyltransferase' and the O-Fuc glycans can get glucuronylated.

Page 12 - note that the CaZy family 105 (TMTC) has members in *Drosophila* and *Caenorhabditis*, but their mannosyltransferase activity is yet to be proven in these organisms.

Table 1: probably no O-glycan in *Drosophila* carries sialic acid and for O-Man, only a mannose can be confirmed and the tetrasaccharide for O-Fuc is also wrong. The spelling of 'Gogi' (i.e., Golgi) needs correction; remove word 'body' from the various 'functions', correct 'abd' to 'and'. For O-Man 'growth and embryonic/organ development' should be enough. In the case of O-Glc, xylosylation by 'shams' is known. The PE containing glycolipid in *Drosophila* has a GalNAc on the GlcNAc - recheck with Seppo/Tiemeyer paper.

GlcA on O-glycans has been examined at a genetic level by Tiemeyer/Nishihara.

Page 13 - write 'there are' not 'there're'

Page 15 - wheatgerm agglutinin can be used for O-GlcNAc

Page 16 - 'kurodai' with small 'k'

Page 17: the glycolipid termin quoted are incorrect and must be re-checked with the literature - e.g., GalNAc-alpha1,4-GalNAc-beta1,4- as far as I know. PE is present on dipteran glycolipids and AEP on those of molluscs.

Figure 3: is the first structure with AEP for *Drosophila* really correct? in which paper was this published? Spelling of 'Caeno' not 'Carno'.

Pages 19-21 - recently novel GAG structures were found in a parasitic nematode worm and relatively long GAG chains released by hydrazinolysis of *C. elegans*. Sulphation is at a low level for *C. elegans* CS.

Page 21: the authors ask about the reasons for glycosylation - as Varki wrote in one review, no cell exists without surface sugars. Sugars confer heterogeneity and mean you have many different forms of a single glycoprotein which mean you can tune interactions and activities.

Decision letter (RSOB-18-0232.R0)

17-Dec-2018

Dear Dr Zhu,

We are writing to inform you that the Editor has reached a decision on your manuscript RSOB-18-0232 entitled "Structures and functions of invertebrate glycosylation", submitted to Open Biology.

As you will see from the reviewers' comments below, there are a number of criticisms that prevent us from accepting your manuscript at this stage. The reviewers suggest, however, that a revised version could be acceptable, if you are able to address their concerns. If you think that you can deal satisfactorily with the reviewer's suggestions, we would be pleased to consider a revised manuscript.

The revision will be re-reviewed, where possible, by the original referees. As such, please submit the revised version of your manuscript within six weeks. If you do not think you will be able to meet this date please let us know immediately.

When submitting your revised manuscript, please respond to the comments made by the referee(s) and upload a file "Response to Referees" in "Section 6 - File Upload". You can use this to document any changes you make to the original manuscript. In order to expedite the processing of the revised manuscript, please be as specific as possible in your response to the referee(s).

Please see our detailed instructions for revision requirements
<https://royalsociety.org/journals/authors/author-guidelines/>

Sincerely,

The Open Biology Team
 mailto: openbiology@royalsociety.org

Reviewer(s)' Comments to Author(s):

Referee: 1

Comments to the Author(s)

As stated by the authors, the rich diversity of invertebrate species, genomes, and the time and cost efficiency of raising and experimenting on these species have enabled a handful of the species to become excellent model organisms.

This is an interesting review and the background, the objectives of the work have been properly described. Before accept, there are still some points that I want to discuss with the authors to make the review be clearly addressed as below.

1. Page 2, line 19-22, "Studying invertebrate glycosylation, especially... assists understanding glycobiology and targeted glycoengineering in both invertebrates and vertebrates." What is the most differences between the invertebrates and vertebrates in the glycobiology and targeted glycoengineering? Please add the most concise summary to this part.

2. Page 4, Figure 1, Examples of N-glycans in invertebrate species, I can not find the relative description of this figure in the text. Please position it.

3. Page 13, Table 1, also I can not find the description in the text. Please position it in the text.

4. Page 21-22, Concluding remarks The conclusion is so long which will make the paper not clear and comprehensive, so please refine the conclusion carefully.

Referee: 2

Comments to the Author(s)

Open Biology, Zhu et al.

The authors have sought to summarise data on glycans in invertebrates, especially *Drosophila* and *Caenorhabditis*. However, there are some inaccuracies in this review which must be corrected regardless of the final publication decision for Open Biology - I will not claim to have found all mistakes and grammatical errors. I also do not seek that it be comprehensive, but it must be accurate.

Page 1, Line 5: The first sentence of the abstract is one of the most common statements about glycosylation - the exact same phrase can be found in Ceciliani et al, 2007, in Hayes et al, 2012 or Cornelissen et al, 2016 (to name the first few google 'hits'). The authors should seek to have at least the first sentence different to those elsewhere in the literature.

Line 18: 'on the model organismS' (plural), then spelling of *Caenorhabditis* (not *Carno* ...)

Page 2, Line 6: "97%" would be of animal species not of all the earth's species - put a semi-colon (;) after the word 'invertebrates'.

Line 8: I would not say that the 'vast majority' of invertebrate glycomic studies have been on recombinant proteins - certainly a large number, but what about individual glycoprotein antigens from *Echinococcus*, glycoprotein allergens from insect venoms or haemocyanins from snails, never mind the various whole-organism studies?

Page 3, line 15 - not 'inside the body of mammals', rather 'inside animal cells'.
 line 18: 'terminal' rather than 'end'
 line 21: 'glycoSYLtransferases'

Figure 1: panel A is ok, but for panel B - *Trichomonas* is not an invertebrate species (note spelling of species) - it is a protist; the authors could put a PC on the structure in *Lymantria* (PC also found in *Trichoplusia ni*), add a trifucosylated core for *C. elegans* or a GalFuc core with methylaminoethylphosphonate from a marine snail, a xylosylated glycan from some species (e.g., *Schistosoma* or snail) or triantennary PE-modified glucuronylated glycans from royal jelly (recent paper in MCP). They should also add mannose to the set of monosaccharides. AEP is aminoethylphosphONate.

Page 4, line 13 - write 'insect cells' not 'insect body'.

Page 5, line 8 - what is meant by a high degree of structure conservation with those in mammals? which structures? Of course, three antennae are possible in *C. elegans*, but these are not galactosylated - the galactose residues being on fucose residues or on mannose (e.g., bisecting position).

line 11 - write 'dipteran species, specifically mosquito larvae, were ...'

line 15 - PE is phosphoethanolamine; *Volvarina* (Eckmair, MCP, 2016) is a mollusc and does not fit so well in the same sentence as two moths - but could be mentioned elsewhere - with branched Fuc, GlcA modifications, phosphonate and a trace of PC.

line 23 - 'glycoSYLtransferases'

Page 6, line 15 - a number of glycomics studies have been done on gene knock-out strains in *C. elegans* - e.g., *fut-1/fut-6/fut-8*, *gly-12/gly-13/gly-14*, *hex-2/hex-3*, *aman-2*, *bre-1*. RNAi has also been done on glycogenes from *C. elegans*. Also the *nac* mutant (GDP-Fuc) transporter in *Drosophila* was investigated glycomically (Geisler, JBC).

Page 7, line 4 - ribose is not found in invertebrate N-glycans, but GalNAc is; check spelling of N-glycoylneuraminic. Further variation with methyl groups or anionic/zwitterionic groups could be mentioned.

Page 7, line 16 and pages 8/9 - the routine nature might be 'large scale N-glycoproteomics' to identify glycosylation sites, but the in-depth analysis to find new structures in invertebrates is far from routine and certainly Glycoworkbench is not ok for dealing with novel structures! The 'analytical' discussion focuses on glycoproteomics and does not mention the great advances recently made with off-line LC-MALDI-MS in order to find new structures.

Page 9, line 16 - suggest 'O-glycosylation in invertebrates' (also without word 'the')

Page 10, line 11 - 'however, few of these have been ...'

Figure 2: the first structure shown might well be a breakdown product of a GAG; structures from *Drosophila* with GlcA could also be drawn, based on the literature - far from everything is in UnicarbKB!

Page 11, line 5 - actually if looking at Kurz et al, Figure 13 and Suppl Figure 11, then it is shown that *Drosophila* has sulphated O-glycans; *Vespula* has PE (not PC) on its O-glycans.

line 13 - spelling of Golgi (not 'Gogi')

line 23 - fringe is not a 'glucotransferase' but an 'N-acetylglucosaminyltransferase' and the O-Fuc glycans can get glucuronylated.

Page 12 - note that the CaZy family 105 (TMTC) has members in *Drosophila* and *Caenorhabditis*, but their mannosyltransferase activity is yet to be proven in these organisms.

Table 1: probably no O-glycan in *Drosophila* carries sialic acid and for O-Man, only a mannose can be confirmed and the tetrasaccharide for O-Fuc is also wrong. The spelling of 'Gogi' (i.e., Golgi) needs correction; remove word 'body' from the various 'functions', correct 'abd' to 'and'. For O-Man 'growth and embryonic/organ development' should be enough. In the case of O-Glc, xylosylation by 'shams' is known. The PE containing glycolipid in *Drosophila* has a GalNAc on the GlcNAc - recheck with Seppo/Tiemeyer paper. GlcA on O-glycans has been examined at a genetic level by Tiemeyer/Nishihara.

Page 13 - write 'there are' not 'there're'

Page 15 - wheatgerm agglutinin can be used for O-GlcNAc

Page 16 - 'kurodai' with small 'k'

Page 17: the glycolipid termin quoted are incorrect and must be re-checked with the literature - e.g., GalNAc-alpha1,4-GalNAc-beta1,4- as far as I know. PE is present on dipteran glycolipids and AEP on those of molluscs.

Figure 3: is the first structure with AEP for *Drosophila* really correct? in which paper was this published? Spelling of 'Caeno' not 'Carno'.

Pages 19-21 - recently novel GAG structures were found in a parasitic nematode worm and relatively long GAG chains released by hydrazinolysis of *C. elegans*. Sulphation is at a low level for *C. elegans* CS.

Page 21: the authors ask about the reasons for glycosylation - as Varki wrote in one review, no cell exists without surface sugars. Sugars confer heterogeneity and mean you have many different forms of a single glycoprotein which mean you can tune interactions and activities.

Author's Response to Decision Letter for (RSOB-18-0232.R0)

See Appendix A.

RSOB-18-0232.R1 (Revision)

Review form: Reviewer 2

Recommendation

Accept with minor revision (please list in comments)

Are each of the following suitable for general readers?

- a) **Title**
Yes
- b) **Summary**
Yes
- c) **Introduction**
Yes

Is the length of the paper justified?

Yes

Should the paper be seen by a specialist statistical reviewer?

No

Is it clear how to make all supporting data available?

Not Applicable

Is the supplementary material necessary; and if so is it adequate and clear?

Not Applicable

Do you have any ethical concerns with this paper?

No

Comments to the Author

Open Biology, Zhu et al., revised

The authors have revised their summary of data on glycans in invertebrates, especially *Drosophila* and *Caenorhabditis*. However, there are some smaller points for revision.

Page 1, Line 18: not corrected - 'on the model organismS' (plural), then spelling of *Caenorhabditis* (not *Carno* ...)

Page 2, Line 3: 'occurS' - singular form of verb with 's' on end.

Page 4, Line 23: 'phosphorYLcholine' (also page 6, line 22; page 16, line 6; page 18, line 9)

Page 5, Line 1: 'phosphoethANOLamine' (also page 10, line 21 and page 16, line 2)

Page 10, Line 15: 'addS' - singular form of verb with 's' on end.

Page 10, Line 22: italicise *Vespula germanica*

Page 12, Line 8: the introduced part of the sentence is not correct, as Ichimaya et al did do in vitro assays on co-transfected Sf21 cells to show activity. What is meant rather is that the TMTC-type mannosyltransferases of CaZy family 105, not the POMT1/2 ones, have not yet been assayed in *Caenorhabditis* & *Drosophila*. TMTC is an alternative O-mannosylation pathway relevant to cadherins rather than dystroglycan.

Page 13, Line 6: 'O-GlcNAcylation can actually occur in the nucleus and cytosol'

Page 13, Line 14 - at end of paragraph add: "There is also an extracellular form of O-GlcNAcylation on EGF repeats mediated by the EOGT enzyme in the ER." (e.g., review by Varshney & Stanley, 2017, *Biochem Soc Trans*).

Page 15, Line 9 - 'Glycosphingolipids in invertebrates'

Page 15, Line 23: "In *Drosophila*, the core mactosyl Man(b1-4)Glc structure"

Page 16, Line 3 - '...phosphoNate'

Page 16, Line 7 - "GSLs of the lepidopteran species *Bombyx mori* were ... but novel extensions were revealed." (italicise *Bombyx mori*)

Page 16, Line 10 - as mactosyl ceramide is defined on page 15, no need to define again.

Page 17, Line 10 - 'Glycosaminoglycans in invertebrates'

Page 17, Line 18 - "only heparan and chondroitin chains with or without sulphate."

Page 17, Lines 19/20 - the sentence starting 'Invertebrates' repeats the one at the end of the previous paragraph and so can be deleted.

Page 18, Line 8 - "heparan, 91, whereby the relevant enzymes synthesising the core are encoded by genes defective in sqv mutants" (here refer to Esko/Horvitz)

Page 18, Line 18 - remove the sentences starting 'for example' and 'the GAGs' - instead write "As for *Caenorhabditis*, the GAGs in *Drosophila* are based on the same canonical GlcA-Gal-Gal-Xyl core for attachment to proteins; some of the relevant enzymes have been characterised, such as oxt (Wilson, J Biol Chem, 2002) and GalT7 (Vadaie & Jarvis, J Biol Chem, 2002).

Page 19 - line 18 - can also add 'or hydrazinolysis'

Ref 41: 'Guérardel' (not with capital ÉR)

Ref 82: not J Biol Chem but J Biochem (Tokyo)

Ref 84: italicise *Drosophila* and remove >

Table 1: For O-Fuc, no Gal in *Drosophila*, rather a branched trisaccharide GlcA(GlcNAc)Fuc; also O-Man is probably not extended. Suggest making all glycans scaled to same size of monosaccharide.

Figure 1: the Volvarina structure is methylated on the unknown antennal hexose.

Figure 2: First three *Drosophila* structures, the terminal HexNAc or Hex are undefined (not necessarily GlcNAc or Gal); the sixth structure should have no terminal Gal; the seventh/last structure are fine if the undefined HexNAc would be sulphated.

Decision letter (RSOB-18-0232.R1)

07-Jan-2019

Dear Dr Zhu

We are pleased to inform you that your manuscript RSOB-18-0232.R1 entitled "Structures and functions of invertebrate glycosylation" has been accepted by the Editor for publication in *Open Biology*. The reviewer(s) have recommended publication, but also suggest some minor revisions to your manuscript. Therefore, we invite you to respond to the reviewer(s)' comments and revise your manuscript.

Please submit the revised version of your manuscript within 14 days. If you do not think you will be able to meet this date please let us know immediately and we can extend this deadline for you.

- 1) A text file of the manuscript (doc, txt, rtf or tex), including the references, tables (including captions) and figure captions. Please remove any tracked changes from the text before submission. PDF files are not an accepted format for the "Main Document".
- 2) A separate electronic file of each figure (tiff, EPS or print-quality PDF preferred). The format should be produced directly from original creation package, or original software format. Please note that PowerPoint files are not accepted.
- 3) Electronic supplementary material: this should be contained in a separate file from the main text and meet our ESM criteria (see <http://royalsocietypublishing.org/instructions-authors#question5>). All supplementary materials accompanying an accepted article will be treated as in their final form. They will be published alongside the paper on the journal website and posted on the online figshare repository. Files on figshare will be made available approximately one week before the accompanying article so that the supplementary material can be attributed a unique DOI.

Online supplementary material will also carry the title and description provided during submission, so please ensure these are accurate and informative. Note that the Royal Society will not edit or typeset supplementary material and it will be hosted as provided. Please ensure that the supplementary material includes the paper details (authors, title, journal name, article DOI). Your article DOI will be 10.1098/rsob.2016[last 4 digits of e.g. 10.1098/rsob.20160049].

- 4) A media summary: a short non-technical summary (up to 100 words) of the key findings/importance of your manuscript. Please try to write in simple English, avoid jargon, explain the importance of the topic, outline the main implications and describe why this topic is newsworthy.

Images

Data-Sharing

It is a condition of publication that data supporting your paper are made available. Data should be made available either in the electronic supplementary material or through an appropriate

repository. Details of how to access data should be included in your paper. Please see <http://royalsocietypublishing.org/site/authors/policy.xhtml#question6> for more details.

Data accessibility section

Sincerely,

The Open Biology Team
<mailto:openbiology@royalsociety.org>

Reviewer(s)' Comments to Author:

Referee: 2

Comments to the Author(s)
 Open Biology, Zhu et al., revised

The authors have revised their summary of data on glycans in invertebrates, especially *Drosophila* and *Caenorhabditis*. However, there are some smaller points for revision.

Page 1, Line 18: not corrected - 'on the model organismS' (plural), then spelling of *Caenorhabditis* (not *Carno* ...)

Page 2, Line 3: 'occurS' - singular form of verb with 's' on end.

Page 4, Line 23: 'phosphorYLcholine' (also page 6, line 22; page 16, line 6; page 18, line 9)

Page 5, Line 1: 'phosphoethANOLamine' (also page 10, line 21 and page 16, line 2)

Page 10, Line 15: 'addS' - singular form of verb with 's' on end.

Page 10, Line 22: italicise *Vespula germanica*

Page 12, Line 8: the introduced part of the sentence is not correct, as Ichimaya et al did do in vitro assays on co-transfected Sf21 cells to show activity. What is meant rather is that the TMTC-type mannosyltransferases of CaZy family 105, not the POMT1/2 ones, have not yet been assayed in *Caenorhabditis* & *Drosophila*. TMTC is an alternative O-mannosylation pathway relevant to cadherins rather than dystroglycan.

Page 13, Line 6: 'O-GlcNAcylation can actually occur in the nucleus and cytosol'

Page 13, Line 14 - at end of paragraph add: 'There is also an extracellular form of O-GlcNAcylation on EGF repeats mediated by the EOGT enzyme in the ER.' (e.g., review by Varshney & Stanley, 2017, Biochem Soc Trans).

Page 15, Line 9 - 'Glycosphingolipids in invertebrates'

Page 15, Line 23: "In *Drosophila*, the core mactosyl Man(b1-4)Glc structure"

Page 16, Line 3 - '...phosphoNate'

Page 16, Line 7 - "GSLs of the lepidopteran species *Bombyx mori* were ... but novel extensions were revealed." (italicise *Bombyx mori*)

Page 16, Line 10 - as mactosyl ceramide is defined on page 15, no need to define again.

Page 17, Line 10 - 'Glycosaminoglycans in invertebrates'

Page 17, Line 18 - "only heparan and chondroitin chains with or without sulphate."

Page 17, Lines 19/20 - the sentence starting 'Invertebrates' repeats the one at the end of the previous paragraph and so can be deleted.

Page 18, Line 8 - "heparan, 91, whereby the relevant enzymes synthesising the core are encoded by genes defective in sqv mutants" (here refer to Esko/Horvitz)

Page 18, Line 18 - remove the sentences starting 'for example' and 'the GAGs' - instead write "As for *Caenorhabditis*, the GAGs in *Drosophila* are based on the same canonical GlcA-Gal-Gal-Xyl core for attachment to proteins; some of the relevant enzymes have been characterised, such as oxt (Wilson, J Biol Chem, 2002) and GalT7 (Vadaie & Jarvis, J Biol Chem, 2002).

Page 19 - line 18 - can also add 'or hydrazinolysis'

Ref 41: 'Guérardel' (not with capital ÉR)

Ref 82: not J Biol Chem but J Biochem (Tokyo)

Ref 84: italicise *Drosophila* and remove

Table 1: For O-Fuc, no Gal in *Drosophila*, rather a branched trisaccharide GlcA(GlcNAc)Fuc; also O-Man is probably not extended. Suggest making all glycans scaled to same size of monosaccharide.

Figure 1: the Volvarina structure is methylated on the unknown antennal hexose.

Figure 2: First three *Drosophila* structures, the terminal HexNAc or Hex are undefined (not necessarily GlcNAc or Gal); the sixth structure should have no terminal Gal; the seventh/last structure are fine if the undefined HexNAc would be sulphated.

Author's Response to Decision Letter for (RSOB-18-0232.R1)

See Appendix B.

Decision letter (RSOB-18-0232.R2)

08-Jan-2019

Dear Dr Zhu

We are pleased to inform you that your manuscript entitled "Structures and functions of invertebrate glycosylation" has been accepted by the Editor for publication in Open Biology.

Sincerely,

The Open Biology Team
mailto: openbiology@royalsociety.org

Appendix A

Reviewer(s)' Comments to Author(s):

Referee: 1

Comments to the Author(s)

As stated by the authors, the rich diversity of invertebrate species, genomes, and the time and cost efficiency of raising and experimenting on these species have enabled a handful of the species to become excellent model organisms.

This is an interesting review and the background, the objectives of the work have been properly described. Before accept, there are still some points that I want to discuss with the authors to make the review be clearly addressed as below.

1. Page 2, line 19-22, "Studying invertebrate glycosylation, especially... assists understanding glycobiology and targeted glycoengineering in both invertebrates and vertebrates." What is the most differences between the invertebrates and vertebrates in the glycobiology and targeted glycoengineering? Please add the most concise summary to this part.

Response: The N-glycosylation pathway in mammals is clear (Page 3 Line 17-22), however, invertebrates' N-glycosylation pathway have not been fully elucidated and remains controversial (Page 4 Line 1-8). Glycan structures are species specific and can alter upon physiological, pathological, or developmental stages for both invertebrates and vertebrates. Invertebrate glycans was once thought not as complicated as in vertebrates, but increasing studies have been demonstrating the unexpected rich diversity of invertebrate glycans.

2. Page 4, Figure 1, Examples of N-glycans in invertebrate sepecies, I can not find the relative description of this fiugre in the text. Please position it.

Response: We have added discussion on Figure 1 and referenced Figure 1 in the manuscript (see text).

3. Page 13, Table 1, also I can not find the description in the text. Please position it in the text.

Response: We thank the reviewer for the comments and have added discussion on Table 1 and referenced it in the text (see text).

4. Page 21-22, Concluding remarks The conclusion is so long which will make the paper not clear and comprehensive, so please refine the conclusion carefully.

Response: We have shortened and refined the concluding remarks so that it is more concise and comprehensive (see text).

Referee: 2

Comments to the Author(s)

Open Biology, Zhu et al.

The authors have sought to summarise data on glycans in invertebrates, especially *Drosophila* and *Caenorhabditis*. However, there are some inaccuracies in this review which must be corrected regardless of the final publication decision for Open Biology - I will not claim to have found all mistakes and grammatical errors. I also do not seek that it be comprehensive, but it must be accurate.

Response: We thank the reviewer for the valuable comments and we have carefully revised the manuscript to minimize grammatical errors and ensure accurate and appropriate expressions. Changes are highlighted in red.

Page 1, Line 5: The first sentence of the abstract is one of the most common statements about glycosylation - the exact same phrase can be found in Ceciliani et al, 2007, in Hayes et al, 2012 or Cornelissen et al, 2016 (to name the first few google 'hits'). The authors should seek to have at least the first sentence different to those elsewhere in the literature.

Response: We thank the reviewer for the suggestion, and have modified the first sentence so that it does not resemble other papers.

“Glycosylation refers to the covalent attachment of sugar residues to a protein or lipid, and the biological importance of this modification has been widely recognized.”

Line 18: 'on the model organismS' (plural), then spelling of Caenorhabditis (not Carno ...)

Response: We thank the reviewer for the corrections and has corrected in the manuscript.

Page 2, Line 6: "97%" would be of animal species not of all the earth's species - put a semi-colon (;) after the word 'invertebrates'.

Response: We have made the corrections in the manuscript.

Line 8: I would not say that the 'vast majority' of invertebrate glycomic studies have been on recombinant proteins - certainly a large number, but what about individual glycoprotein antigens from Echinococcus, glycoprotein allergens from insect venoms or haemocyanins from snails, never mind the various whole-organism studies?

Response: We appreciate the reviewer's comments and have modified the sentence in the text.

"So far, many invertebrate glycomic studies have focused on recombinant glycoproteins, for example, expressed using the baculovirus system. Recent years, increasing numbers of studies have been focusing on the glycomes originally derived from invertebrate species."

Page 3, line 15 - not 'inside the body of mammals', rather 'inside animal cells'.

Response: Correction has been made.

line 18: 'terminal' rather than 'end'

Response: Correction has been made.

line 21: 'glycoSYLtransferases'

Response: We thank the reviewer for the correction and have revised the term throughout the text.

Figure 1: panel A is ok, but for panel B - Trichomonas is not an invertebrate species (note spelling of species) - it is a protist; the authors could put a PC on the structure in Lymantria (PC also found in Trichoplusia ni), add a

trifucosylated core for *C. elegans* or a GalFuc core with methylaminoethylphosphonate from a marine snail, a xylosylated glycan from some species (e.g., *Schistosoma* or snail) or triantennary PE-modified glucuronylated glycans from royal jelly (recent paper in MCP). They should also add mannose to the set of monosaccharides. AEP is aminoethylphosphONate.

Response: We thank the reviewer for the valuable comments. We have added more N-glycan structures from additional invertebrate species in Figure 1 as suggested by the reviewer, and have deleted the structure from the protest *Trichomonas*. Mannose have been added to the monosaccharide set in the figure legend.

Page 4, line 13 - write 'insect cells' not 'insect body'.

Response: Correction has been made.

Page 5, line 8 - what is meant by a high degree of structure conservation with those in mammals?

which structures? Of course, three antennae are possible in *C. elegans*, but these are not galactosylated - the galactose residues being on fucose residues or on mannose (e.g., bisecting position).

Response: We thank the reviewer for the suggestion and have modified the sentence to avoid overstatement.

"...the nematode Caenorhabditis also contains a nearly contiguous series of N-glycans"

line 11 - write 'dipteran species, specifically mosquito larvae, were ...'

Response: Corrections have been made.

line 15 - PE is phosphoethanolamine; Volvarina (Eckmair, MCP, 2016) is a mollusc and does not fit so well in the same sentence as two moths - but could be mentioned elsewhere - with branched Fuc, GlcA modifications, phosphonate and a trace of PC.

Response: We thank the reviewer for the correction. The sentence have been modified in the text.

“...core difucosylated and zwitterion phosphorycholine and phosphoethylamine modified N-glycans were also identified in a handful of invertebrate species such as Trichoplusia ni and Lymantria dispar. The mollusc Volvarina rubella was also found to contain N-glycans with phosphonate and phosphorcholine modifications in addition to Fuc and GlcA modifications. Additionally, xylosylated glycans and triantennary phosphoethylamine-modified glucuronylated glycans have also been identified from Schistosoma and honeybee royal jelly, respectively.”

line 23 - ‘glycoSYLtransferases’

Response: correction has been made.

Page 6, line 15 - a number of glycomics studies have been done on gene knock-out strains in *C. elegans* - e.g., fut-1/fut-6/fut-8, gly-12/gly-13/gly-14, hex-2/hex-3, aman-2, bre-1. RNAi has also been done on glycogenes from *C. elegans*. Also the nac mutant (GDP-Fuc) transporter in *Drosophila* was investigated glycomically (Geisler, JBC).

Response: We thank the reviewer for the valuable comments. We have added in the text the discussion regarding the genetically engineered invertebrate species for understanding the functions of specific glycoenzymes as well as relevant literature.

“In fact, genetically engineered mutants of Caenorhabditis and Drosophila have been established in order to reveal the biological functions of specific glycoenzymes such as fucosyltransferases, N-acetylglucosaminyltransferases, hexosaminidases, and other glycoenzymes.”

Page 7, line 4 - ribose is not found in invertebrate N-glycans, but GalNAc is; check spelling of N-glycoylneuraminic. Further variation with methyl groups or anionic/zwitterionic groups could be mentioned.

Response: We thank the reviewer for the correction and comments. Spelling has been corrected. Further modification of the monosaccharides has been discussed in the text.

“Further modifications of the monosaccharides such as methylation, sulphation, and zwitterionic modification including phosphorycholine, phosphoethanolamine and aminoethyl phosphate have also been identified in invertebrates.”

Page 7, line 16 and pages 8/9 - the routine nature might be ‘large scale N-glycoproteomics’ to identify glycosylation sites, but the in-depth analysis to find new structures in invertebrates is far from routine and certainly Glycoworkbench is not ok for dealing with novel structures! The ‘analytical’ discussion focuses on glycoproteomics and does not mention the great advances recently made with off-line LC-MALDI-MS in order to find new structures.

Response: We thank the reviewer for the valuable comments. We agree with the author and have modified the related statements in the text and added discussion on the powerful glycan analysis tool by off-line LC-MALDI-MS.

“Although the filter-aided deamidation method can perform large scale identification of protein glycosylation sites, it cannot provide information on the specific glycan structures attached to the glycosylation site.”

“That said, in-depth analysis of the N-glycomes, especially finding novel glycan structures, can still be quite challenging and far from routine. A combination of exoglycosidase digestion, offline LC separation and purification, as well as MALDI-TOF MS/MS or LC/MS/MS analysis is usually needed in order to reveal new glycan structures.”

Page 9, line 16 - suggest ‘O-glycosylation in invertebrates’ (also without word ‘the’)

Response: We have revised the section title to “O-glycosylation in invertebrates”.

Page 10, line 11 - ‘however, few of these have been ...’

Response: We have modified the sentence.

Figure 2: the first structure shown might well be a breakdown product of a GAG; structures from *Drosophila* with GlcA could also be drawn, based on the literature - far from everything is in UnicarbKB!

Response: We thank the reviewer for the correction and suggestion. We have modified Figure 2 in the manuscript to include *Drosophila* mucin type O-glycans.

Page 11, line 5 - actually if looking at Kurz et al, Figure 13 and Suppl Figure 11, then it is shown that *Drosophila* has sulphated O-glycans; *Vespula* has PE (not PC) on its O-glycans.

Response: We thank the reviewer for the correction, and has revised the sentence in the text.

“So far sulfated O-glycans have been identified in Drosophila but not Caenorhabditis, and phosphoethylamine modification has been reported in the insect Vespula germanica...”

line 13 - spelling of Golgi (not ‘Gogi’)

Response: Correction has been made.

line 23 - fringe is not a ‘glucotransferase’ but an ‘N-acetylglucosaminyltransferase’ and the O-Fuc glycans can get glucuronylated.

Response: We thank the reviewer for the correction. Correction has been made and we have modified the sentence in the manuscript (see text).

Page 12 - note that the CaZy family 105 (TMTC) has members in *Drosophila* and *Caenorhabditis*, but their mannosyltransferase activity is yet to be proven in these organisms.

Response: We thank the reviewer for the valuable comments, and have modified the sentence in the text.

“... POMT1 and POMT2, function in association with each other to maintain normal muscle development, though their mannosyltransferase activity in this organism is yet to be proven.”

Table 1: probably no O-glycan in *Drosophila* carries sialic acid and for O-Man, only a mannose can be confirmed and the tetrasaccharide for O-Fuc is also wrong. The spelling of ‘Gogi’ (i.e., Golgi) needs correction; remove word ‘body’ from the various ‘functions’, correct ‘abd’ to ‘and’. For O-Man ‘growth and

embryonic/organ development' should be enough. In the case of O-Glc, xylosylation by 'shams' is known. The PE containing glycolipid in *Drosophila* has a GalNAc on the GlcNAc - recheck with Seppo/Tiemeyer paper. GlcA on O-glycans has been examined at a genetic level by Tiemeyer/Nishihara. **Response:** We thank the reviewer for the comments and correction. The spelling mistakes have been corrected. The word "body" has been removed from various "functions". The O-Man functions have been modified. Xylosylation by 'shams' have been added to the discussion. The structures of the exemplary glycans in Table 1 have been redrawn to ensure consistence with literature (see text).

Page 13 - write 'there are' not 'there're'

Response: We have modified the sentence in the text.

Page 15 - wheatgerm agglutinin can be used for O-GlcNAc

Response: We agree with the reviewer that wheat germ agglutinin, specifically, lectin weak affinity chromatography, can be applied for O-GlcNAc enrichment and have incorporated in the discussion (see text).

Page 16 - 'kurodai' with small 'k'

Response: Correction has been made.

Page 17: the glycolipid termin quoted are incorrect and must be re-checked with the literature - e.g., GalNAc- α 1,4-GalNAc- β 1,4- as far as I know. PE is present on dipteran glycolipids and AEP on those of molluscs.

Response: We thank the reviewer for the correction and comments. We have corrected and revised the sentence in the text.

"In Drosophila, the core structure can be further modified with a GalNAc β 1-4-GlcNAc β 1-3 residue, and the terminal Gal can be further capped with GlcA. Phosphoethylamine is present as a typical modification to dipteran glycolipids and aminoethylphosphoate to those of molluscs."

Figure 3: is the first structure with AEP for *Drosophila* really correct? in which paper was this published? Spelling of 'Caeno' not 'Carno'.

Response: We apologize for the drawing mistake and have corrected the structures in Figure 3

The spelling has also been corrected.

Pages 19-21 - recently novel GAG structures were found in a parasitic nematode worm and relatively long GAG chains released by hydrazinolysis of *C. elegans*. Sulphation is at a low level for *C. elegans* CS.

Response: We thank the reviewer for the valuable comments and we have modified the discussion on the GAG structures in nematodes and sulfation level in the text.

“The common tetrasaccharide core linking the repeating disaccharides and the serine residue of the proteoglycan are reported as GlcA(β1-3)Gal(β1-3)Gal(β1-4)Xyl for chondroitin and heparan. Recently, novel GAG tetrasaccharide core with additional galactose and phosphocholine modifications were reported for the parasitic nematode Oesophagostomum dentatum. ”

Page 21: the authors ask about the reasons for glycosylation - as Varki wrote in one review, no cell exists without surface sugars. Sugars confer heterogeneity and mean you have many different forms of a single glycoprotein which mean you can tune interactions and activities.

Response: We thank the reviewer for the valuable comments. We have revised the sentence in the text.

“Glycosylation confers heterogeneity on glycoconjugates and finely tunes their structures and functions.”

Appendix B

The authors have revised their summary of data on glycans in invertebrates, especially *Drosophila* and *Caenorhabditis*. However, there are some smaller points for revision.

Page 1, Line 18: not corrected - ‘on the model organismS’ (plural), then spelling of *Caenorhabditis* (not *Carno* ...)

Response: We are sorry that we missed the correction. Now the corrections have been corrected.

Page 2, Line 3: ‘occurS’ - singular form of verb with ‘s’ on end.

Response: Correction has been made.

Page 4, Line 23: ‘phosphorYLcholine’ (also page 6, line 22; page 16, line 6; page 18, line 9)

Response: We thank the reviewer for the correction. Correction has been made throughout.

Page 5, Line 1: ‘phosphoethANOLamine’ (also page 10, line 21 and page 16, line 2)

Response: We thank the reviewer for the correction. Correction has been made throughout.

Page 10, Line 15: ‘addS’ - singular form of verb with ‘s’ on end.

Response: Correction has been made

Page 10, Line 22: italicise *Vespula germanica*

Response: Correction has been made

Page 12, Line 8: the introduced part of the sentence is not correct, as Ichimaya et al did do in vitro assays on co-transfected Sf21 cells to show activity. What is meant rather is that the TMTC-type mannosyltransferases of CaZy family 105, not the POMT1/2 ones, have not yet been assayed in *Caenorhabditis* & *Drosophila*. TMTC is an alternative O-mannosylation pathway relevant to cadherins rather than dystroglycan.

Response: We thank the reviewer for the correction. We have modified the sentence in the text.

*“Another type of O-mannosylation, such as for the cadherin superfamily, depends on the TMTC-type mannosyltransferases for O-mannosylation, however, their mannosyltransferase activity in *Caenorhabditis* and *Drosophila* is yet to be proven.”*

Page 13, Line 6: 'O-GlcNAcylation can actually occur in the nucleus and cytosol'

Response: We have revised the sentence.

Page 13, Line 14 - at end of paragraph add: 'There is also an extracellular form of O-GlcNAcylation on EGF repeats mediated by the EOGT enzyme in the ER.' (e.g., review by Varshney & Stanley, 2017, Biochem Soc Trans).

Response: We thank the reviewer for the suggestion. We have added the sentence to the end of the graph and cited the relevant literature.

Page 15, Line 9 - 'Glycosphingolipids in invertebrates'

Response: Revision has been made.

Page 15, Line 23: "In Drosophila, the core mactosyl Man(b1-4)Glc structure"

Response: We have revised the sentence according to the reviewer's suggestion.

Page 16, Line 3 - '...phosphoNate'

Response: Correction has been made.

Page 16, Line 7 - "GSLs of the lepidopteran species *Bombyx mori* were ... but novel extensions were revealed." (italicise *Bomby mori*)

Response: The sentence has been revised based on the reviewer's suggestion.

Page 16, Line 10 - as mactosyl ceramide is defined on page 15, no need to define again.

Response: We have deleted the redundant description of the mactosyl ceramide.

Page 17, Line 10 - 'Glycosaminoglycans in invertebrates'

Response: We have revised the section title.

Page 17, Line 18 - "only heparan and chondroitin chains with or without sulphate."

Response: The sentence has been revised according to the reviewer's suggestion.

Page 17, Lines 19/20 - the sentence starting 'Invertebrates' repeats the one at the end of the previous paragraph and so can be deleted.

Response: We have deleted the sentence.

Page 18, Line 8 - "heparan, 91, whereby the relevant enzymes synthesising the core are encoded by genes defective in sqv mutants" (here refer to Esko/Horvitz)

Response: We thank the reviewer for the valuable comments. We have added the sentence and cited the relevant paper.

Page 18, Line 18 - remove the sentences starting 'for example' and 'the GAGs' - instead write "As for Caenorhabditis, the GAGs in Drosophila are based on the same canonical GlcA-Gal-Gal-Xyl core for attachment to proteins; some of the relevant enzymes have been characterised, such as oxt (Wilson, J Biol Chem , 2002) and GalT7 (Vadaie & Jarvis, J Biol Chem, 2002).

Response: We thank the reviewer for the suggestion. The original sentences have been deleted and replaced by the sentence suggested by the reviewer.

Page 19 - line 18 - can also add 'or hydrazinolysis'

Response: The method "hydrazinolysis" has been added to the sentence.

Ref 41: 'Guérardel' (not with capital ÉR)

Response: Correction has been made.

Ref 82: not J Biol Chem but J Biochem (Tokyo)

Response: We thank the reviewer for the correction. Correction has been made.

**Ref 84: italicise Drosophila and remove **

Response: correction has been made.

Table 1: For O-Fuc, no Gal in Drosophila, rather a branched trisaccharide GlcA(GlcNAc)Fuc; also O-Man is probably not extended. Suggest making all glycans scaled to same size of monosaccharide.

Response: The O-Fuc and O-Man glycan structures in Table 1 have been modified according to the reviewer's suggestion. The glycans have been sized roughly to the same scale.

Figure 1: the Volvarina structure is methylated on the unknown antennal hexose.

Response: We thank the reviewer for the correction and have modified the glycan structure in Figure 1.

Figure 2: First three Drosophila structures, the terminal HexNAc or Hex are undefined (not necessarily GlcNAc or Gal); the sixth structure should have no terminal Gal; the seventh/last structure are fine if the undefined HexNAc would be sulphated.

Response: The Drosophila mucin type O-glycans in Figure 2 are from reference 38, which have provided monosaccharide identity for these structures. To be more inclusive, we replaced the terminal GlcNAc or Gal by HexNAc or Hex as suggested by the reviewer.